# Understanding In Vitro Pathways to Drug Discovery for TDP-43 Proteinopathies

**DOI:** 10.3390/ijms232314769

**Published:** 2022-11-25

**Authors:** Hei W. A. Cheng, Timothy B. Callis, Andrew P. Montgomery, Jonathan J. Danon, William T. Jorgensen, Yazi D. Ke, Lars M. Ittner, Eryn L. Werry, Michael Kassiou

**Affiliations:** 1School of Medical Sciences, Faculty of Medicine and Health, The University of Sydney, Sydney, NSW 2006, Australia; 2School of Chemistry, Faculty of Science, The University of Sydney, Sydney, NSW 2006, Australia; 3Department of Biomedical Sciences, Faculty of Medicine and Health Sciences, 2 Technology Place, Macquarie University, Sydney, NSW 2109, Australia; 4Central Clinical School, Faculty of Medicine and Health, The University of Sydney, Sydney, NSW 2006, Australia

**Keywords:** frontotemporal dementia, amyotrophic lateral sclerosis, TDP-43, proteinopathy, stress granules

## Abstract

The use of cellular models is a common means to investigate the potency of therapeutics in pre-clinical drug discovery. However, there is currently no consensus on which model most accurately replicates key aspects of amyotrophic lateral sclerosis (ALS) and frontotemporal dementia (FTD) pathology, such as accumulation of insoluble, cytoplasmic transactive response DNA-binding protein (TDP-43) and the formation of insoluble stress granules. Given this, we characterised two TDP-43 proteinopathy cellular models that were based on different aetiologies of disease. The first was a sodium arsenite-induced chronic oxidative stress model and the second expressed a disease-relevant TDP-43 mutation (TDP-43 M337V). The sodium arsenite model displayed most aspects of TDP-43, stress granule and ubiquitin pathology seen in human ALS/FTD donor tissue, whereas the mutant cell line only modelled some aspects. When these two cellular models were exposed to small molecule chemical probes, different effects were observed across the two models. For example, a previously disclosed sulfonamide compound decreased cytoplasmic TDP-43 and increased soluble levels of stress granule marker TIA-1 in the cellular stress model without impacting these levels in the mutant cell line. This study highlights the challenges of using cellular models in lead development during drug discovery for ALS and FTD and reinforces the need to perform assessments of novel therapeutics across a variety of cell lines and aetiological models.

## 1. Introduction

Proteinopathy relating to the transactive response DNA-binding protein 43 (TDP-43) is a pathological hallmark in almost all cases of amyotrophic lateral sclerosis (ALS) [1] and approximately 50% of frontotemporal dementia (FTD) patients [2]. Postmortem brain tissue exhibits mislocalisation of TDP-43 from the nucleus to the cytoplasm, where it undergoes ubiquitination, phosphorylation and forms aggregates [1]. Nuclear depletion of TDP-43 results in an inability of TDP-43 to perform its normal physiological functions such as facilitating alternative splicing and mRNA transcription [3]. Progression of disease has been linked to loss-of-function from nuclear clearance of TDP-43 and gain-of-toxicity through cytotoxic effects, although the mechanisms involved are not well understood.

One pathomechanism proposed to be implicated in TDP-43 proteinopathy is the formation of stress granules. Stress granules are membraneless cytoplasmic mRNA-protein granules formed under conditions of cellular stress, such as oxidative stress or heat shock [4,5]. TDP-43, along with other RNA-binding proteins such as FUS and T-cell intracytoplasmic antigen (TIA-1) are consolidated into stress granules, where they are believed to facilitate cell survival via the storage of non-essential RNA-binding proteins during stress exposure [6]. Mutations in the TDP-43 gene are linked to an increased incidence of ALS and produce a form of TDP-43 with a higher tendency to colocalise with stress granules [7].

In vitro models of acute oxidative stress, such as exposure to the oxidative phosphorylation uncoupler sodium arsenite, induce reactive oxygen species and free radicals that cause a temporary accumulation of TDP-43-positive stress granules. These stress granules quickly dissipate after removal of the stressor, highlighting their dynamic nature [8]. Under chronic stress exposure, however, liquid–liquid phase separation occurs and RNA granules with low complexity domains, such as those containing TDP-43, undergo concentration-dependent phase transition to hydrogel states [9]. These hydrogels retain their dynamic properties, but will mature over time with continued stressor exposure to form pathological inclusions which are insoluble in nature [10]. The application of chronic oxidative stressors also induces several other features consistent with TDP-43 proteinopathies, such as a loss of nuclear TDP-43 and accumulation of diffuse TDP-43 in the cytosol [11,12]. In mouse models of ALS, the reduction of stress granule-associated proteins can rescue degenerative phenotypes [13]. Given there are no treatment options that reverse pathophysiology in ALS and FTD, the distinct link between TDP-43 pathology and stress granules in the progression of ALS and FTD suggest both TDP-43 mistranslocation and stress granules could be suitable targets for developing disease-modifying treatments [14].

In the typical ALS and FTD drug discovery process, high-throughput screens targeting TDP-43 aggregation, cleavage or motor neuron survival have been used as an initial read-out of activity [15,16]. These screens are often performed on different cellular models and under different experimental conditions, which may exhibit varied pathological characteristics. For example, HeLa cells treated with the ER stressor thapsigargin (1 mM, 50 min) showed colocalisation of TDP-43 with stress granules, whereas Neuro2a cells treated with a higher dose of thapsigargin (5 mM, 4 h) did not exhibit colocalisation [17,18]. Furthermore, certain stressors used to stimulate stress granule formation may upregulate downstream proteins, whereas others do not. Hydrogen peroxide is a commonly used stressor in cellular models and can stimulate stress granule formation in U2OS cells (2 mM, 1 h) [19]. In induced pluripotent stem cell (iPSC) models, however, only sodium arsenite and heat shock, and not hydrogen peroxide (2 mM, 1 h), stimulated stress granule formation. Sodium arsenite and heat shock both selectively upregulated the phosphorylation of stress granule proteins such as eukaryotic translation initiation factor 2 (eIF2α) [20]. To date, there is no consensus in the TDP-43 proteinopathy field about which in vitro assay or TDP-43 model best mimics the disease pathology. This understanding would assist in assessing the validity of previous models used in the literature and the resultant drugs optimised using those models. This also raises the question of whether more than one assay is required to identify lead compounds in high-throughput screens.

Given this, we conducted an in-depth characterisation of two different common cellular models of ALS and FTD. The first was an oxidative stress model using HEK-293 cells overexpressing TDP-43 WT treated with a chronic dose of sodium arsenite. The second used SH-SY5Y neuroblastoma cells stably expressing a disease-relevant mutant form of TDP-43 (TDP-43 M337V). We used Western blots to assess whether TDP-43, stress granule and ubiquitin localisation and solubility in these models mirrored that seen in human ALS and FTD. The sodium arsenite model recapitulated most aspects of TDP-43, stress granule and ubiquitin pathology, whereas the mutant model only displayed some aspects. As this may suggest different aetiologies vary in their pathology, both models were used to test the ability of two small molecules to correct aberrant changes.

## 2. Results and Discussion

### 2.1. Establishing a TDP-43 WT-Overexpressing HEK-293 Cell Line—An Oxidative Stress Model

HEK-293 cells produce endogenous TDP-43, but in order to induce overexpression of TDP-43, the Flp-In™ system was used [21]. This recombinase-based system uses a cytomegalovirus promoter to drive a tetracycline-dependent overexpression of TDP-43 WT protein.

Once established, cells were exposed to tetracycline (0.5, 1 and 2 µg/mL) for various time periods (24, 48 h) to determine optimal TDP-43 overexpression levels. Treatment with 2 μg/mL of tetracycline for 48 h was sufficient to induce overexpression of TDP-43 in the cytoplasm and nucleus, as well as an increase in both soluble and insoluble TDP-43, as compared to endogenous TDP-43 produced in HEK-293 cells (Figure 1). It was not, however, sufficient to induce increased stress granule formation, indicated by no change in TIA-1 in tetracycline-treated HEK-293 cells compared to untreated HEK-293 cells (Figure 2A–D).

To better model stress granule formation, tetracycline-treated TDP-43-overexpressing cells were stressed with sodium arsenite. Sodium arsenite is an oxidative stressor which generates reactive oxidative species and free radicals [22]. A number of studies examining the role of stress granules in TDP-43 proteinopathy have opted for an acute, high-dosage treatment of sodium arsenite between 0.25–5 mM for 0.5–1 h [12,23]. However, it has been shown that a mild chronic dose of sodium arsenite, such as 15 μM for 18 h, induces cellular changes that better mimic those seen in TDP-43 proteinopathies [12]. In addition, studies on fibroblasts have shown colocalisation of TIA-1 with stress granules after mild, chronic treatment of sodium arsenite, but not after an acute dose [12].

Given this precedent, we opted to establish an oxidative stress model by treating tetracycline (2 μg/mL, 48 h)-induced TDP-43-over-expressing cells with a mild, chronic dose of sodium arsenite (18 h, 15 μM, Figure 1C–F). This increased the expression levels of both monomeric TDP-43 (43 kDa) and a C-terminal fragment of 35 kDa (CTF35) in the nucleus (Figure 1D), both of which are found in TDP-43 inclusions isolated from postmortem ubiquitin-positive FTD (FTLD-U) brains [1]. Furthermore, sodium arsenite also significantly increased cytoplasmic TDP-43 levels in comparison to untreated HEK-293 cells; however, levels did not rise as high as tetracycline-treated levels. This indicates that overexpression alone can induce sufficient TDP-43 mislocalisation (Figure 1C). In addition, treatment with sodium arsenite globally increased both RIPA-soluble and urea-soluble TDP-43 levels (Figure 1E,F).

Sodium arsenite is also commonly used to induce stress granule formation [21,22]. Overexpression of TDP-43 by addition of tetracycline did not impact levels of TIA-1 in the nucleus or the cytoplasm (Figure 2A,B), confirming previous findings with a similar HEK TDP-43 Flp-In model [24]. However, when sodium arsenite was added to the TDP-43-overexpressing HEK-293 cells, levels of the stress granule marker, TIA-1, significantly increased in the cytoplasm and nucleus (Figure 2A,B). This global increase in TIA-1 is also reflected in an increase in RIPA-soluble and urea-soluble TIA-1 (Figure 2C,D). Immunofluorescence reinforced these results from Western blotting, showing that cytoplasmic mislocalisation of TDP-43 and TIA-1 was primarily observed after sodium arsenite treatment and not tetracycline alone (Figure 3).

The changes induced by this chronic, mild sodium arsenite treatment of HEK-293 TDP-43-over-expressing cells mirror many aspects seen in human postmortem FTD and/or ALS tissue. Increased cytoplasmic aggregation of TDP-43 is seen in FTD and ALS tissue [25,26], as well as in our sodium arsenite model. The stress granule marker TIA-1 appears in the cytoplasm and nucleus of neurons and glia from the spinal cord and motor cortex of human ALS donor tissue, but is either entirely absent [27] or found only in the nucleus of neurons and glia from control tissue [23,28,29]. The sodium arsenite model mimics this increase in cytoplasmic localisation seen in human tissue, and the more contentious increase in nuclear overexpression. Increased TIA-1 is observed in both soluble and insoluble fractions of lumbar spinal cord from a SOD1 G86S patient, as well as from iPSC-derived neural stem cells established from a SOD1 G17S ALS patient [28], as also seen in the sodium arsenite model. Increased TIA-1 is observed in both soluble and insoluble fractions of lumbar spinal cord from a SOD1 G86S patient, as well as from iPSC-derived neural stem cells established from a SOD1 G17S ALS patient [28], as also seen in the sodium arsenite model. Furthermore, ubiquitinated inclusions are seen in both the cytoplasm and nucleus of FTD/ALS cases [30], as shown in our sodium arsenite model.

The one aspect of human ALS/FTD pathology that was not observed in our sodium arsenite model related to nuclear TDP-43 levels. Sodium arsenite induced an increase in nuclear TDP-43, whereas nuclear TDP-43 levels measured in postmortem ALS tissue decreased compared to controls [26]. That being said, the degree of nuclear TDP-43 staining in FTDL-U cases varied widely [25], so it is possible that the sodium arsenite model more closely recapitulates nuclear TDP-43 levels seen in a subset of FTD cases that displayed nuclear TDP-43 inclusions rather than in ALS.

In summary, these TIA-1-related features of human ALS pathology are an improvement on previous models that fail to demonstrate an increase in cytoplasmic and nuclear TIA-1. Human neuroblastoma BE-M17 cells overexpressing TDP-43-EGFP show TIA-1 mainly in the nucleus, and only upregulate TIA-1 expression in the cytoplasm on arsenite exposure [23]. Heat shock of colorectal carcinoma RKO cells does not alter TIA-1 distribution in the cytoplasm or nucleus [31]. In contrast, SOD1 G93A mutant mice and NSC-34 cells transfected with mutant SOD-1 show an increase in cytoplasmic TIA-1 localisation with a decrease in nuclear localisation [28]. This highlights how sensitive TIA-1 pathology is to the type of stressors that induce TDP-43 proteinopathy or cell types subjected to that stress.

### 2.2. Establishing a SH-SY5Y TDP-43 M337V Mutant Cellular Model

An aspect not accounted for in the TDP-43 WT-overexpressing HEK-293 cell line is the presence of disease-relevant TDP-43 mutants such as M337V observed in a portion of familial ALS patients [8] To this end, we characterised the TDP-43, TIA-1 and ubiquitin distribution in a cell model expressing this mutant compared to wild-type TDP-43. We used neuroblastoma SH-SY5Y cells that were stably transfected with TDP-43 M337V (V5-tagged). The V5-tag allowed us to differentiate the effect of drugs specifically on the TDP-43 M337V mutant as compared to the endogenous TDP-43 expressed in SH-SY5Y cells without compromising the function of the protein. TDP-43 M337V expression did not affect the nuclear or cytoplasmic localisation of TDP-43 (Figure 4A,B), as also shown in other studies [8,32,33]. The over-expression of M337V TDP-43 also suppressed expression of endogenous TDP-43 in the SH-SY5Y TDP-43 M337V mutant cellular model (Figure 4B), which is consistent with previous reports [34]. The expression level of RIPA-soluble TDP-43 M337V decreased, whereas urea-soluble TDP-43 increased, as compared to TDP-43 WT in SH-SHY5Y cells (Figure 4C,D).

Immunofluorescence imaging also shows similar expression levels of TDP-43 and TIA-1 in SH-SY5Y cells transfected with TDP-43 M337V as compared to untransfected SH-SY5Y cells (Figure 5, Figure 6). In addition, this imaging reveals the appearance of aggregates in TDP-43 M337V cells, reinforcing the Western blot data that suggest a redistribution from soluble to insoluble TDP-43 stores.

### 2.3. Rationale of Compound ***1*** and ***2***

To examine whether small molecules are able to rescue the molecular and cellular changes seen in the two characterised models, a hit compound from a previous high-throughput screen of approximately 75,000 small molecules was selected (**1**, Figure 7) [35]. In that study, compound **1** reduced whole cell GFP expression in sodium arsenite-treated (15 μM, 18 h) rat PC-12 cells transfected with TDP-43:GFP under control of a tetracycline-off system. This effect was dose dependent, with an IC_50_ of approximately 300 nM. Western blots on whole-cell soluble and insoluble fractions of PC-12 cells treated with **1** showed a global reduction in monomeric and higher molecular weight TDP-43:GFP in insoluble fractions [36]. Comparison of compound **1**’s shape and electrostatic properties to an inhouse library, allowed us to identify compound **2**, which had similar features. We ran this novel chemotype, compound **2**, alongside **1** in our characterised assays, examining their ability to impact TDP-43, stress granule and ubiquitin solubility and localisation.

As can be seen, **1** and **2** had no effect on TDP-43 expression in the SH-SY5Y TDP-43 M337V mutant cellular model, but **1** reduced cytoplasmic TDP-43 in the oxidative stress model. Compound **1** locally reduced TDP-43 levels in the cytoplasm of HEK-293 TDP-43 cells treated with sodium arsenite, without altering nuclear levels of TDP-43 (Figure 8A,B) or levels of TDP-43 in soluble and insoluble fractions. This suggests the reduction of cytoplasmic TDP-43 by compound **1** could be induced through mechanisms that do not involve changing the soluble and insoluble ratio of TDP-43 (Figure 8C,D). Compound **2** had no effect, despite retaining similar electrostatic properties to compound **1**. Significant structural changes, such as substituting a sulfonamide for an ether linker, appears to impact the ability of these compounds to alter TDP-43 cell pathology.

After SH-SY5Y TDP-43 M337V cells were treated with either compounds **1** or **2**, no statistically significant changes to the nucleocytoplasmic distribution of TDP-43 M337V were observed (Figure 8E,F), although this may have been expected given the SH-SY5Y TDP-43 M337V model did not show any altered nucleocytoplasmic TDP-43 distribution compared to WT SH- SY5Y cells. Compounds **1** and **2**, however, did not correct the decrease in RIPA-soluble TDP-43 M337V displayed in this model, nor did they impact the increase in urea-soluble TDP-43 M337V (Figure 8G,H). From these results, we can conclude that these compounds do not interact with the TDP-43 M337V variant in a manner that affects the TDP-43 distribution.

The lack of effect of compound **1** on insoluble TDP-43 in the HEK-293 oxidative stress model conflicts with a previous study [35], which found that compound **1** induced a significant reduction in soluble and insoluble monomeric and aggregated TDP-43 in whole cell lysates of PC-12 cells overexpressing TDP-43-GFP stressed with sodium arsenite. Measures were taken to replicate such results, such as using the IC_50_ (300 nM) of **1** as determined in Boyd et al. in the present study. Furthermore, our HEK-293 oxidative stress model was stressed with the same mild, chronic dose of sodium arsenite (18 h, 15 uM) as Boyd in their high-throughput screen. However, when they performed Western blots to determine the effect of **1** on insoluble TDP-43, cells were acutely treated with a high dose of sodium arsenite (0.5 mM, 1 h). It is possible that different doses of sodium arsenite induce different cell morphology and alter TDP-43 recruitment ability into stress granules; therefore, the different oxidative stress intensities between the studies may have contributed to the difference in findings. Other discrepancies that may have contributed include the difference in TDP-43 type (wild type in the present study vs GFP-tagged in the Boyd et al. paper) and the cellular background (HEK-293 in the present study vs PC-12 cells in the Boyd paper). This may further highlight the need to perform the same assays across multiple cell lines to build an accurate picture of the effect of drugs on TDP-43 in vitro.

### 2.4. ***1*** and ***2*** Alter Stress Granule Solubility in Both Cellular Models

Compound **1** had no effect on the nucleocytoplasmic distribution of TIA-1 in TDP-43 WT HEK-293 cells (Figure 9A) whereas compound **2** significantly decreased nuclear TIA-1 expression, with no effect on cytoplasmic TIA-1 (Figure 9B). Both compounds **1** and **2** increased RIPA-soluble TIA-1 and reduced urea-soluble TIA-1 (Figure 9C,D) in the TDP-43 HEK-293 oxidative stress model. In the SH-SY5Y TDP-43 M337V mutant model, urea-soluble TIA-1 was significantly reduced after treatment with compound **1**, whereas RIPA-soluble TIA-1 levels were not impacted. Compound **2,** however, reduced both RIPA-soluble and microaggregated urea-soluble TIA-1 (Figure 9G,H). This suggests the compounds impart an alteration of the solubility profile of stress granules from insoluble to soluble. Soluble stress granules are more easily degraded and are able to perform their normal function as a translational silencer, whereas insoluble stress granules are microaggregated and exhibit more prion-like characteristics [36,37], suggesting these compounds move stress granules into a more favourable state.

The nucleocytoplasmic distribution of TIA-1 in the SH-SY5Y TDP-43 M337V cells was not significantly affected after treatment with compounds 1 and 2. (Figure 9E,F).

Given the similar effects of compounds 1 and 2 on insoluble TIA-1 levels in both cellular models, the pathway by which these molecules work to alter the insoluble fraction of TIA-1 may be the same in both cellular models, although the mechanism of their effects on the soluble fraction of TIA-1 may be different. We speculate that compounds **1** and **2** may selectively promote clearance of insoluble TIA-1.

### 2.5. Downregulation of Ubiquitin in Both Cellular Models

Previous studies have shown that inhibition of the ubiquitin–proteasome system is linked to cytoplasmic stress granule formation [38]. Ubiquitination of stress granule marker G3BP1 can also mediate stress granule disassembly, although the mechanism of disassembly is dependent on how the stress was initiated [39]. Given this, we investigated whether compound-induced changes to stress granule solubility were associated with increased activity in the ubiquitin–proteasome system. Treatment with compound **2** resulted in a reduction in cellular ubiquitin levels in the TDP-43 WT HEK-293 oxidative stress model, whereas treatment with compound **1** did not alter the ubiquitin expression levels as compared to control (Figure 9I,J). Downregulation of ubiquitination was observed in the SH-SY5Y TDP-43 M337V mutant cellular model after treatment with **1** and **2** (Figure 9K,L). This suggests that neither compound stimulated the recruitment of the ubiquitin–proteasome system, and that the inhibition of the ubiquitin–proteasome system seen here did not link to an increase in cytoplasmic stress granule formation as suggested by other studies [38].

### 2.6. Viability

It is unclear whether these effects of compound **1** and **2** on the biochemical profile of TDP-43, stress granules and ubiquitination would yield useful outcomes on cell viability. It could be predicted that a shift of stress granules from insoluble to soluble states may be useful for cells [40,41]; however, it is not clear whether this impact is enough to improve viability of cells. In addition, the ability of compound **1** to decrease soluble TDP-43 in the HEK-293 oxidative stress model may have a deleterious effect by reducing the amount of soluble TDP-43 that can participate in physiological functions such as RNA regulation and transcription. Given this complexity, we wanted to investigate whether compounds **1** and **2** would alter cell viability (Figure 10). MG-132, a proteasome inhibitor known to induce cellular toxicity [42,43], was used as a positive control.

MG-132 significantly impaired cell survival in both cell models (Figure 10). Neither compound **1** nor **2** protected sodium arsenite-treated HEK-293 cells or SH-SY5Y TDP-43 M337V cells from cell death, nor did they induce cell death. The cell viability data suggest the shift in solubility profile of stress granules from insoluble to soluble imparted by **1** and **2** is not toxic to cells but is not protective either. It also suggests that reducing soluble TDP-43 in the HEK-293 cell model does not impact viability.

## 3. Materials and Methods

### 3.1. General Procedures

The synthetic routes and synthetic procedures of compounds **1** and **2** are outlined in Appendix A.

### 3.2. Reagents

Unless otherwise stated, all chemicals were purchased from Sigma-Aldrich (St. Louis, MO, USA).

### 3.3. SH-SY5Y Cell Culture

Human neuroblastoma SH-SY5Y cells stably transfected with TDP-43 M337V were kindly provided by Lars Ittner (Macquarie University, Sydney). Cells were cultured in Dulbecco’s modified Eagle’s medium: Nutrient Mixture F-12 (DMEM/F12) with 10% fetal bovine serum (Thermo Fisher Scientific, Waltham, MA, USA), 1% penicillin-streptomycin and 25 μg/mL blasticidin S hydrochloride. Cells were grown at 37 °C in 5% CO_2_.

### 3.4. Generation of Wild-Type TDP-43-Expressing HEK-293 Cells

HEK-293 cells stably expressing human wild-type TDP-43 (TDP-43 WT HEK-293 cells) were established using the Flp-In™ system (Life Technologies, Carlsbad, CA, USA). Briefly, a pcDNATM5/FRT/TO plasmid containing human TDP-43 cDNA and a pOG44 plasmid (Life Technologies) were propagated in endonuclease and recombinase-deficient competent *E. coli* (BIO-85027, Bioline, London, UK) by adding 50 ng of plasmid to competent *E. coli* and incubating on ice for 30 min. Cells were heat shocked at 42 °C for 45 s then incubated with Luria–Bertani (LB) media for 1 h. Cultures were then incubated for 16 h at 37 °C on LB agar plates (Life Technologies) containing 100 μg/mL ampicillin. Antibiotic-resistant positive transformants were inoculated into LB broth. Both pcDNA5/FRT/TO and pOG44 plasmids were purified using the Plasmid Midiprep System (Promega, Madison, WI, USA) according to manufacturer’s protocols. Generation of wild-type TDP-43-expressing HEK-293 cells.

Flp-In™ T-REx™ HEK293 cells (Life Technologies) were cultured in Dulbecco’s modified Eagle’s medium (Thermo Fisher Scientific) with 10% fetal bovine serum, 1% penicillin-streptomycin, 15 μg/mL blasticidin S hydrochloride and 100 μg/mL zeocin selection reagent (Life Technologies). Stable transfections were performed using FuGENE HD^®^ transfection reagent (Promega) according to the manufacturer’s protocol by co-transfecting pcDNA5 plasmid containing wild-type TDP-43 cDNA sequence and pOG44 plasmid containing Flp-recombinase cDNA at a 1:10 ratio. Hygromycin (80 μg/mL; Life Technologies) was used to select positive clones. Stably-transfected cells were maintained in Dulbecco’s modified Eagle’s medium with 10% fetal bovine serum, 1% penicillin-streptomycin and 15 μg/mL blasticidin S hydrochloride. The overexpression of TDP-43 was induced by the addition of 2 μg/mL of tetracycline for 48 h prior to protein extraction.

### 3.5. Cellular Treatments

TDP-43 WT HEK-293 cells were treated with sodium arsenite (15 μM) for 18 h, with or without test ligands (300 nM) or MG-132 (5 μM; a proteasome inhibitor used as a positive control). TDP-43 M337V SH-SY5Y cells were only treated with test ligands (300 nM) for 18 h or with 0.1% DMSO as a control. TDP-43 WT HEK-293 cells were grown until 80–90% confluency in a T75 before use and TDP-43 M337V SH-SY5Y cells were grown until 80–90% confluency in a T175 before use.

### 3.6. Nucleocytoplasmic Fractionation

To examine the nucleocytoplasmic distribution of TDP-43, the stress granule marker TIA-1 and ubiquitin, cells were treated with test ligands according to the method described above, washed with warm Dulbecco’s phosphate buffered saline (PBS) and trypsinised with 0.25% trypsin with 0.5 mM EDTA in PBS. Cells were lysed in cold cytoplasmic extraction buffer (10 mM HEPES, 10 mM NaCl, 1 mM KH_2_PO_4_, 5 mM NaHCO_3_, 5 mM EDTA, 1 mM CaCl_2_, 0.5 mM MgCl_2_) with 1X cOmplete EDTA-free protease inhibitors (Roche, Basel, Switzerland) and 1X phosSTOP phosphatase inhibitors (Roche). After incubating on ice for 10 min, cells were sonicated using a probe sonicator at 60% power for 5 s at 4 °C, repeated 5 times. A 2.5 M sucrose solution was added and centrifuged at 6300× *g* for 10 min at 4 °C. The supernatant containing the cytoplasmic fraction was stored at −20 °C. The pellet was resuspended in tris/sucrose/EDTA buffer (10 mM Tris HCl, pH 7.5, 300 mM sucrose, 1 mM EDTA, 0.1% IGEPAL) with 1X cOmplete EDTA-free protease inhibitors and 1X phosSTOP phosphatase inhibitors. The resuspended solution was centrifuged at 1000× *g* for 5 min at 4 °C. The pellet was washed and centrifuged 4 times. The final pellet, corresponding to the nuclear fraction, was resuspended in 100 μL of 2% SDS in cold RIPA buffer and stored in −20 °C until further analysis.

### 3.7. Solubility Fractionation

To examine the effect of drug treatments on the solubility profile of TDP-43 WT HEK-293 cells and TDP-43 M337V SH-SY5Y cells, the RIPA-soluble and urea-soluble fractions were extracted based on established methods [44]. Briefly, after growing in T75 flasks for HEK-293 cells and T175 flasks for SH-SY5Y cells to 80–90% confluency, and treating with drugs as described above, cells were washed with warm Dulbecco’s PBS and trypsinised in trypsin/EDTA. Cells were lysed in cold RIPA supplemented with 1X cOmplete EDTA-free protease inhibitors and 1X phosSTOP phosphatase inhibitors. After incubating on ice for 10 min, cells were sonicated using a probe sonicator at 60% power for 5 s at 4 °C, repeated 5 times. The lysate was centrifuged at 100,000× *g* for 30 min at 4 °C. The supernatant corresponded to the RIPA-soluble fraction. The pellet was washed with cold RIPA buffer by sonicating and centrifuging at 100,000× *g* for 30 min at 4 °C. This was repeated twice, and the supernatant was discarded. The final pellet was sonicated in urea buffer (7 M urea, 2 M thiourea, 4% CHAPS, 0.03 M Tris-HCl, pH 8.5) using a probe sonicator at 60% power for 5 s at 25 °C, repeated 5 times. The resuspended solution was centrifuged at 100,000× *g* for 30 min at 25 °C. The supernatant contained the urea-soluble fraction.

### 3.8. Protein Quantification and Western Blot

The protein concentration of the nuclear, cytosolic, soluble and insoluble fractions was determined using a bicinchoninic acid (BCA) protein assay (Thermo Fisher Scientific) according to the manufacturer’s protocol. Samples (15 μg) for immunoblotting were made up in 2× Laemelli buffer containing 10% β-mecaptoethanol, heat-denatured at 95 °C for 5 min then immediately placed on ice. Protein was electrophoresed on 4–12% Bis-Tris gels (Invitrogen) then dry-transferred onto nitrocellulose (Invitrogen). Blocking was performed by submerging the blots into 5% skim milk in tris-buffered saline with 0.1% Tween-20 (TBS-T) for 1 h. Primary antibodies were made up in either TBS-T or 5% bovine serum albumin (BSA) and incubated at 4 °C overnight. Primary antibodies included rabbit anti-TDP-43 (10782-2-AP, 1:1000, Proteintech, Chicago, IL, USA), rabbit anti-ubiquitin (05-1307, 1:2000, Merck), mouse anti-TIA (sc-48371, 1:500, Santa Cruz Biotechnology, Dallas, TX, USA), mouse anti-tubulin (32–2500, 1:5000, Thermo Fisher Scientific), rabbit anti-lamin B1 (ab16048, 1:5000, Abcam, Cambridge, UK). Blots were washed three times in tri-suffered saline, followed by secondary antibody incubation in TBS-T for 1 h at room temperature. Secondary antibodies include goat anti-rabbit-HRP (A0545, 1:3000) and rabbit anti-mouse-HRP (A9044, 1:5000). The detection of chemiluminescence was performed using enhanced chemiluminescence (ECL) substrate (Thermo Fisher Scientific). Densitometry analyses was performed with ImageLab 6.1 (Bio-Rad Laboratories Inc., Hercules, CA, USA). TDP-43, TIA-1 and ubiquitin band intensities were normalised against α-tubulin for RIPA, urea and cytoplasmic fractions and lamin B1 for nuclear fractions. Data represent mean ± SD of n ≥ 3 repetitions. Significant differences between mean protein expression levels in the different treatment conditions were analysed using GraphPad Prism 8.4.3 software (GraphPad Software, LLC, San Diego, CA, USA) by a one-way ANOVA with Dunnett’s post hoc test. *p* < 0.05 was considered statistically significant.

### 3.9. Immunofluorescence

Cells were grown on poly-D-Lysine-coated 8-well chambered coverglass (Sarstedt, Australia). SH-SY5Y cells (1.5 × 10^3^ cells per well) were plated for 24 h prior to fixing. HEK-293 cells (3 × 10^3^ cells per well) were plated for 24 h prior to treatment with tetracycline (2 μg/mL) for 48 h and sodium arsenite (15 μM) for 18 h. Following treatment, both cell lines were washed twice with warm PBS, then fixed with 4% paraformaldehyde for 10 min. Cells were washed with PBS three times to remove residue paraformaldehyde, each incubating for 5 min. Blocking was performed with 10% donkey serum in PBS for 1 h. Primary antibodies were diluted in BSA solution (1% BSA, 0.2% Tween-20 in PBS) and incubated at 4 °C overnight. The primary antibodies used were rabbit anti-TDP-43 (1:400, Proteintech, Rosemont, IL, USA) and mouse anti TIA-1 (1:200, Santa Cruz, TX, USA). Cells were washed three times in PBS, each incubating for 10 min. Secondary antibodies were diluted in BSA solution and incubated for 1 h, covered from light. The secondary antibodies were donkey anti-rabbit IgG Alexa Fluor 594 (1:200, Invitrogen) and donkey anti-mouse IgG Alexa Fluor 488 (1:200, Invitrogen). Cells were washed three times in PBS, each incubating for 10 min and mounted in Fluoroshield™ with DAPI (Sigma). Microscopy was performed on an ECLIPSE Ni-E fluorescence microscope (Nikon, Tokyo, Japan) carrying lasers 395, 470, 640 nm. Images were captured using a 40× lens. Images were combined into figures using ImageJ and Adobe Illustrator (Adobe, San Jose, CA, USA).

### 3.10. Cell Viability Assay

The CellTitre-Blue^®^ cell viability assay (Promega) was performed to measure cell viability after drug treatments. Both TDP-43 WT HEK-293 cells and TDP-43 M337V SH-SY5Y cells were plated at 4 × 10^4^ cells per well of a 96-well plate. After 24 h, the media was replaced with fresh media with or without drug treatment (300 nM) for 18 h. CellTitre-Blue^®^ (20 μL) was then added and incubated for 4 h at 37 °C. The fluorescence was measured at 560/590 nm using a POLARstar Omega plate reader. Significant differences were analysed with GraphPad Prism 8.4.3 software using a one-way ANOVA with Dunnett’s post hoc test. *p* < 0.05 was considered statistically significant.

## 4. Conclusions

The work in this study characterised two ALS and FTD cellular models commonly used in the literature. Using these models, we demonstrated a previously disclosed compound, **1**, and a novel bioisosteric analogue, **2**, reduce insoluble stress granule levels and ubiquitin expression. These effects do not influence cell viability suggesting **1** and **2** are not toxic to cells. However, they do not reduce stress-induced cell death, suggesting increasing stress granule solubility may not confer cell survival benefits in TDP-43 models.

This study highlights the challenges of using cellular models in lead development during drug discovery for ALS and FTD. We identified differing pathomechanisms in the stress and mutant cellular models, and the conflicting effect of **1** in TDP-43 WT HEK-293 oxidative stress model compared to a previous study in PC-12 cells. At present, there is no cellular model that perfectly demonstrates the pathological changes seen in ALS or FTD. Even motor neurons derived from ALS patient iPSCs vary in their ability to show TDP-43 mislocalisation and aggregation, do not show stress granule formation without exposure to chemical stressors and show line-to-line transcriptomic differences [12,45,46]. It is unclear if this variability between models, both with secondary and iPSC-derived cell lines, reflects differing face validity for ALS and FTD, or actually reflects inter-individual variability in ALS and FTD pathogenesis seen amongst humans. In lieu of a model that best recapitulates ALS and FTD TDP-43-proteinopathy, novel therapeutics should be assessed over a variety of cell lines and aetiological models, rather than one single model, to understand the effectiveness of potential leads and improve the translatability of new drug candidates.

## Figures and Tables

**Figure 1 ijms-23-14769-f001:**
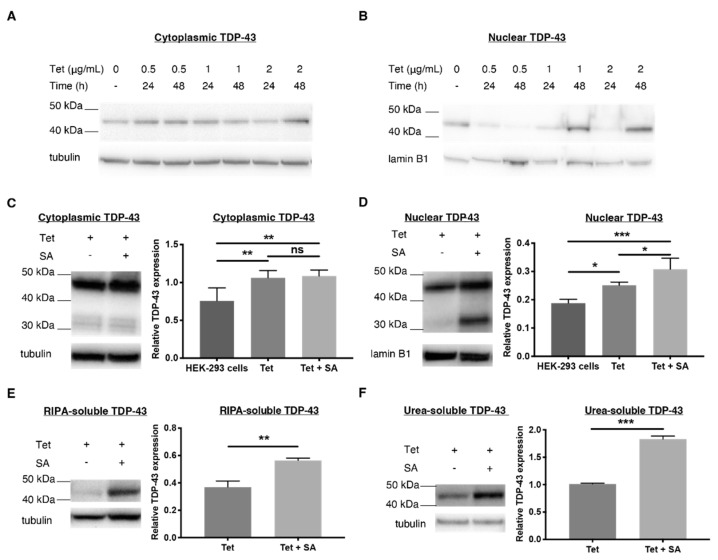
Comparing TDP-43 expression levels in human embryonic kidney (HEK-293) cells stably transfected with tetracycline (Tet)–inducible WT TDP-43 using Western blots. Cells were treated with varied concentrations of Tet (μg/mL) for different time periods. (**A**,**B**) The Western blot shows overexpression of TDP-43 in the cytoplasm and nucleus of transfected HEK-293 cells treated with 2 μg/mL of Tet for 48 h. (**C**,**D**) The quantified expression of TDP-43 in the cytoplasm and nucleus after treatment with Tet (2 μg/mL, 48 h) with and without sodium arsenite (SA, 15 μM, 18 h), an oxidative stressor, to encourage TDP-43 translocation and formation of stress granules. (**E**,**F**) RIPA- and urea-soluble fractions of transfected HEK-293 cells were stained for TDP-43. TDP-43 is normalised to α-tubulin for the cytoplasmic, RIPA-soluble and urea-soluble fractions and lamin B1 for the nuclear fraction. Data represent mean ± SD, n ≥ 3, and significant differences between means were examined with a one-way ANOVA with Tukey’s multiple comparison test. * *p* < 0.05, ** *p* < 0.01, *** *p* < 0.001, ns: not significant.

**Figure 2 ijms-23-14769-f002:**
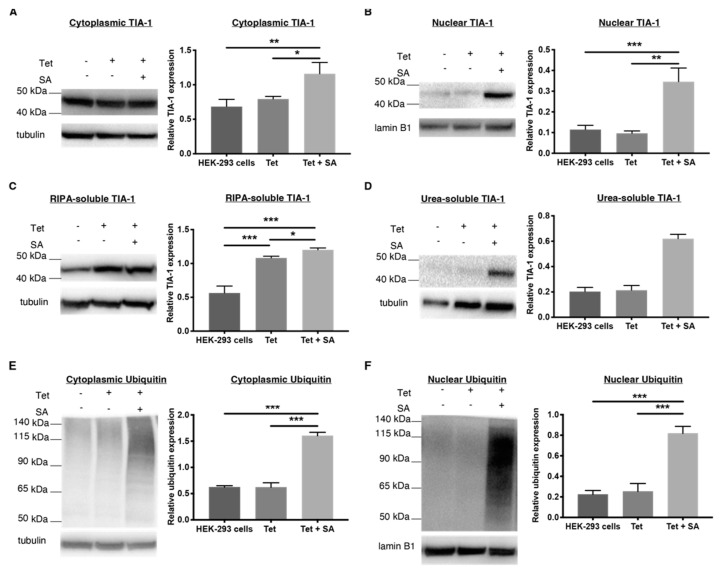
Comparing TIA-1 and ubiquitin expression levels in stably transfected HEK-293 cells. The expression level of the stress granule marker, TIA-1 in stably transfected HEK-293 cells with and without tetracycline (Tet, 2 μg/mL, 48 h) and sodium arsenite (SA, 15 μM, 18 h) was measured using Western blots. (**A**,**B**) The quantified, relative expression levels of TIA-1 in the cytoplasm and nucleus. (**C**,**D**) RIPA- and urea-soluble TIA-1 expression. The Western blots and corresponding histograms showing ubiquitin expression levels in the (**E**,**F**) cytoplasm and nucleus of HEK-293 cells. TIA-1 and ubiquitin are normalised to α-tubulin for the cytoplasmic, RIPA-soluble and urea-soluble fraction and lamin B1 for the nuclear fraction. Data represent mean ± SD, n ≥ 3, and significant differences between means were examined with a one-way ANOVA with Tukey’s multiple comparison test. * *p* < 0.05, ** *p* < 0.01, *** *p* < 0.001.

**Figure 3 ijms-23-14769-f003:**
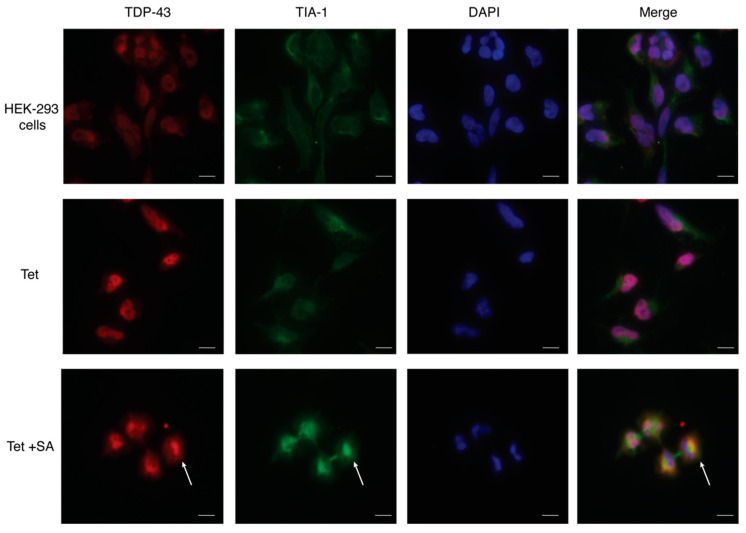
Immunofluorescence imaging of TDP-43 and TIA-1 in stably transfected HEK-293 cells. HEK-293 cells transfected with tetracycline (Tet)–inducible WT TDP-43 (2 ug/mL, 48 h) resulting in overexpression of TDP-43 (red), especially in the nucleus, but low levels of TIA-1. Sodium arsenite (15 μM, 18 h) treated cells displayed mislocalisation of TDP-43 which colocalised with upregulated TIA-1 (green) in the cytoplasm (white arrow). The nuclear stain DAPI (blue) was used to indicate the nucleus. Scale bar = 10 μM.

**Figure 4 ijms-23-14769-f004:**
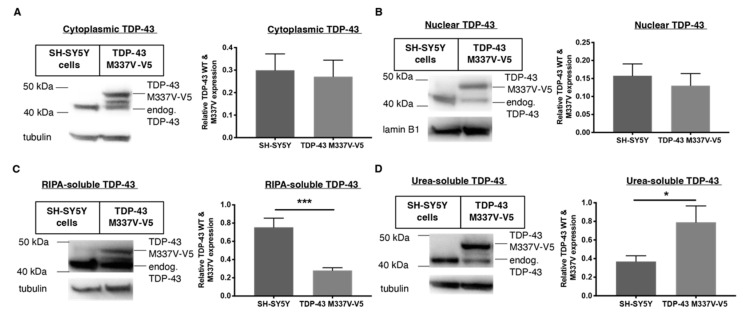
Comparing TDP-43 distribution in human neuroblastoma SH-SY5Y cells stably transfected with TDP-43 M337V mutant (V5-tagged) compared to WT SH-SY5Y cells using Western blots. (**A**,**B**) Cytoplasmic and nuclear TDP-43 expression of SH-SY5Y WT cells compared to transfected SH-SY5Y TDP-43 M337V. (**C**,**D**) Relative RIPA-soluble and urea-soluble expression levels of TDP-43 WT in SH-SY5Y cells compared to TDP-43 M337V mutant. Data represent mean ± SD, n ≥ 3, statistical analysis using parametric, unpaired *t*-test. * *p* < 0.05, *** *p* < 0.001.

**Figure 5 ijms-23-14769-f005:**
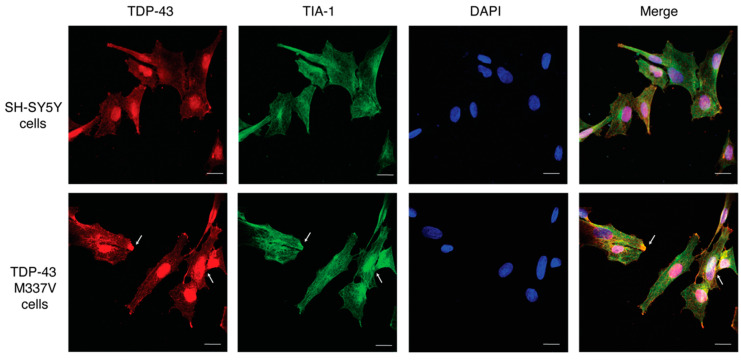
Immunofluorescence imaging of wild–type human neuroblastoma SH-SY5Y cells compared to SH-SY5Y cells expressing the ALS–linked mutation TDP-43 M337V. Representative images of SH-SY5Y cells and M337V TDP-43 SH-SY5Y cells showing similar expression levels and distribution of TDP-43 (red), stress granule marker, and TIA-1 (green). DAPI (blue) was used to indicate the nucleus. The arrow shows the colocalisation of TDP-43 and stress granule aggregates. Scale bar = 10 μM.

**Figure 6 ijms-23-14769-f006:**
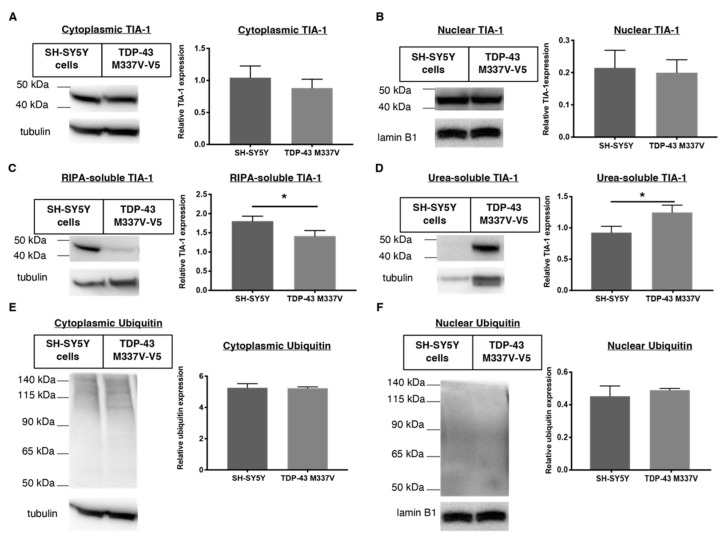
Comparing TIA-1 distribution in wild-type human neuroblastoma SH-SY5Y cells and SH-SY5Y cells stably transfected with TDP-43 M337V mutant (V5-tagged) using Western blots. (**A**,**B**) Cytoplasmic and nuclear TIA-1 expression in SH-SY5Y WT cells compared to transfected SH-SY5Y TDP-43 M337V. The expression of TIA-1 did not significantly change after TDP-43 M337V transfection (**C**,**D**) The solubility of TIA-1 in SH-SY5Y cells compared to TDP-43 M337V mutant. Ubiquitin expression in the (**E**) cytoplasm and (**F**) nucleus. The presence of TDP-43 M337V transfection did not change the level of ubiquitin throughout the cell. Data represent mean ± SD, n ≥ 3, statistical analysis using parametric, unpaired *t*-test. * *p* < 0.05.

**Figure 7 ijms-23-14769-f007:**
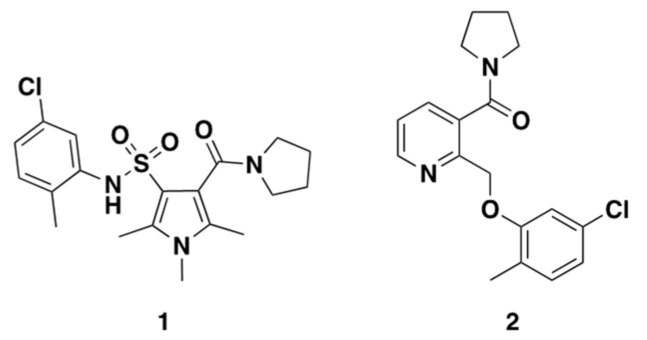
Chemical structures of **1** and **2**.

**Figure 8 ijms-23-14769-f008:**
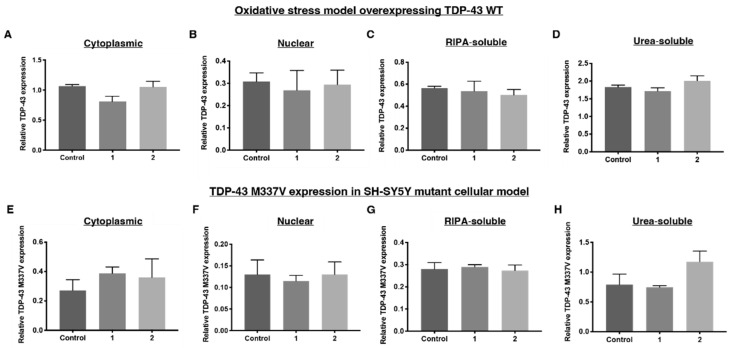
Relative TDP-43 WT expression levels after treatment with 1 and 2 as determined by Western blots. (**A**) Relative TDP-43 WT expression in the cytoplasm, (**B**) nucleus, (**C**) RIPA-soluble fraction and (**D**) urea-soluble fraction after the oxidative stress HEK-293 cellular model, featuring TDP-43 overexpressing HEK-293 cells treated with tetracycline (2 ug/mL, 48 h) and sodium arsenite (15 μM, 18 h), which was exposed to **1** and **2**. Relative TDP-43 M337V-V5 expression levels in the (**E**) cytoplasm, (**F**) nucleus, (**G**) RIPA-soluble fraction and (**H**) urea-soluble fraction from SH-SY5Y TDP-43 M337V mutant cellular model after treatment of **1** and **2**. Both cell lines were treated with **1** and **2** (300 nM) for 18 h. Densitometry of TDP-43 WT and M337V are normalised to α-tubulin for the cytoplasmic, RIPA-soluble and urea-soluble fraction and lamin B1 for the nuclear fraction. Data represent mean ± SD, n ≥ 3. Significant differences were indexed using a one-way ANOVA with Dunnett’s post hoc test.

**Figure 9 ijms-23-14769-f009:**
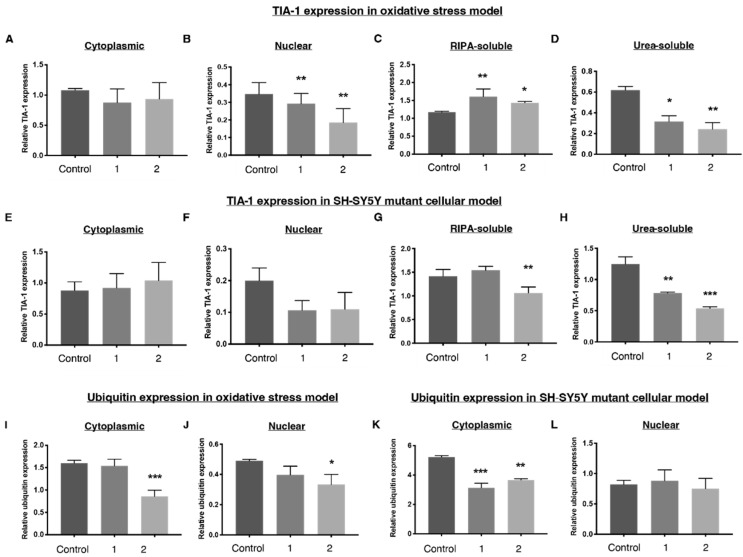
Relative distribution of TIA-1, a stress granule marker and ubiquitin in both cellular models as determined by Western blots. Relative distribution of TIA-1 expression in the (**A**) cytoplasm, (**B**) nucleus, (**C**) RIPA-soluble fraction and (**D**) urea-soluble fraction of the oxidative stress model, featuring HEK-293 cells overexpressing TDP-43 WT, which were treated with tetracycline (2 ug/mL, 48 h) and sodium arsenite (15 μM, 18 h). Relative distribution of TIA-1 in the SH-SY5Y TDP-43 M337V cells in the (**E**) cytoplasm, (**F**) nucleus, (**G**) RIPA-soluble fraction and (**H**) urea-soluble fraction. (**I**,**J**) Ubiquitin expression in the cytoplasm and nucleus of the oxidative stress model overexpressing TDP-43 WT. (**K**,**L**) Ubiquitin expression in the cytoplasm and nucleus of the SH-SY5Y TDP-43 M337V model. Both cell lines were treated with 1 and 2 (300 nM) for 18 h. Densitometry of TIA-1 and ubiquitin are normalised to α-tubulin for the cytoplasmic, RIPA-soluble and urea-soluble fraction and lamin B1 for the nuclear fraction. Data represent mean ± SD, n ≥ 3. Significant differences between means were assessed by a one-way ANOVA with Dunnett’s post hoc test. * *p* < 0.05, ** *p* < 0.01, *** *p* < 0.001.

**Figure 10 ijms-23-14769-f010:**
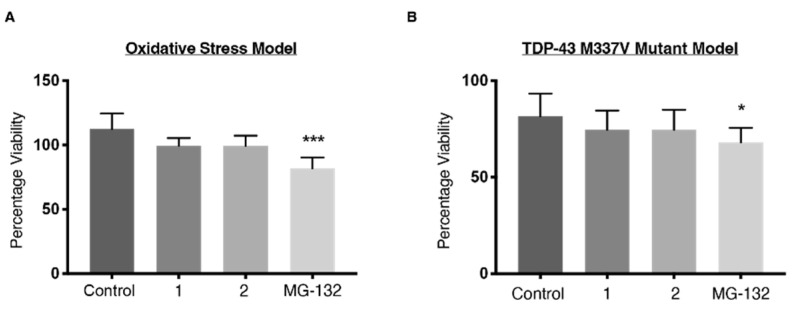
CellTiter-Blue^®^ cell viability assay. (**A**) Results were calculated as a percentage of HEK-293 cells treated with only tetracycline and not sodium arsenite. (**B**) The percentage viability is expressed as a percentage of untreated wild-type SH-SY5Y cells. Both cell lines were treated with **1**, **2** (300 nM) or MG-132 (5 μM) for 18 h. Data represent mean ± SD, n≥ 3. Significant differences were tested using one-way ANOVA with Dunnett’s post-hoc test, * *p* < 0.05, *** *p* < 0.001.

## Data Availability

Not applicable.

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
