# Peer review of "Understanding In Vitro Pathways to Drug Discovery for TDP-43 Proteinopathies"

_ijms, 2022, doi:10.3390/ijms232314769_

Round 1

Reviewer 1 Report

In the present study, Cheng et al. have evaluated two cellular model systems for TDP-43 proteinopathies. Even though the apparent novelty may be low and the outcome may not be very well defined in terms of pathways etc. but the study achieves what the authors set out to do.  As the failures pile up in Neurodegenerative disorders drug discovery programs this study offers an important insight into the follies of drug discovery programs which generally employ a hypothesis-based cellular model. 

Generally, the manuscript is well-organized and carefully written, but I have a few minor suggestions:

1. Figure 1 A: Overexpression is not clearly visible according to the WB for cytoplasmic TDP-43. May want to have a relook. 

2. Figure 8: The title of the top half simply says "TDP-43 WT expression in HEK-293 cell line". The model is actually OS model stressed with Sodium Arsenite. The title shall clearly capture it. Similar issues in Figure 9 and Figure 10.  It will be best to clearly state the OS model characteristics such as exposure time and conc. of sodium arsenite used at some appropriate place.

3. Similar to Figures 1 and 2, Figures 4,6,8, and 9 are also WB based. This shall be clear in the legend itself.

4. If journal style allows perhaps the conclusion should come before Materials and Methods to maintain the flow as there is no separate discussion section. 

Author Response

Response to Reviewer 1:

  1. Figure 1 A: Overexpression is not clearly visible according to the WB for cytoplasmic TDP-43. May want to have a relook. 

We believe Figure 1A reflects overexpression of TDP-43 after treatment of tetracycline, 2 ug/mL for 48 h, compared to untreated cells. This is reinforced in the statistical significance shown in the histogram of 1C. Changes to the figure legend of 1A have been made to clarify the band corresponding to the overexpression.

  1. Figure 8: The title of the top half simply says "TDP-43 WT expression in HEK-293 cell line". The model is actually OS model stressed with Sodium Arsenite. The title shall clearly capture it. Similar issues in Figure 9 and Figure 10.  It will be best to clearly state the OS model characteristics such as exposure time and conc. of sodium arsenite used at some appropriate place.

The suggested changes have been made in the figure, and in the figure legend, for Figures 8, 9 and 10.

  1. Similar to Figures 1 and 2, Figures 4,6,8, and 9 are also WB based. This shall be clear in the legend itself.

Changes have been made in the legends of Figures 1, 2 4, 6, 8 and 9.

  1. If journal style allows perhaps the conclusion should come before Materials and Methods to maintain the flow as there is no separate discussion section. 

If the editors approve this change to the journal style, then we would be happy to move the conclusion up before the Materials and Methods section.

Reviewer 2 Report

It is well known that the accumulation of TDP-43 aggregates in the cells is associated with many neurodegenerative diseases, like amyotrophic lateral sclerosis (ALS), frontotemporal dementia (FTD, etc. Article “Understanding in vitro pathways to drug discovery for TDP-43 proteinopathies” by Cheng et al, is an interesting study to improve our current understanding of the progression of TDP-43-associated proteinopathies. The paper is easy to follow and well explained, all controls have been properly used and the finding is well discussed.  I did not see any major concerns associated with the manuscript. Some minor concerns can be incorporated into the revised version to improve the manuscript further.

1.     Figure 1: Just for clarification: when tubulin was used as a cytoplasmic marker how does it present in urea-solubilized samples as pellet from RIPA solubilization was used for it?

2.     Figure 4A: what is the middle band in M337V-V5 expressing sample?

3.     Figure 4A: what is the middle band in M337V-V5 expressing sample?

Author Response

Response to Reviewer 2:

1.Figure 1: Just for clarification: when tubulin was used as a cytoplasmic marker how does it present in urea-solubilized samples as pellet from RIPA solubilization was used for it?

Tubulin is present in both the RIPA and urea fractions of the cells. The paper below shows that RIPA is able to partially solubilise tubulin proteins and these are able to be detected in the pellet.

http://doi.org/10.1126/scisignal.2005966

There is also precedent within the literature of using tubulin as a loading control for RIPA and urea-soluble fractions within SH-SY5Y cells.

https://doi.org/10.1007/s12035-014-8644-6

  1. Figure 4A: what is the middle band in M337V-V5 expressing sample?

We believe the middle band is a posttranslationally modified version of TDP-43. It is possible that the middle band represents phosphorylated TDP-43, which is 45 kDa. Further testing would be required to confirm this and is outside the scope of this article.  As a control, we had stained gels with a V5 antibody to confirm the location of the M337V-V5 band relative to the ladder.

Round 2

Reviewer 1 Report

All changes ok, may be accepted for publication.

Reviewer 2 Report

In the revised version, authors have addressed all my concerns raised in previous version.
